# Communication and relationship satisfaction of Fly-in, Fly-out workers and partners

Ashlee Forshaw[1], Kristie-Lee Alfrey[1], Corneel Vandelanotte[1], Benjamin Gardner[2], Amanda L. Rebar[1,3]*

1 Motivation of Health Behaviours Lab; Appleton Institute; School of Health, Medical, and Applied Sciences; Central Queensland University; Rockhampton, Australia, 2 School of Psychology, University of Surrey, Guildford, United Kingdom, 3 Motivation of Health Behaviors Lab; Health Promotion, Education, & Behavior; Arnold School of Public Health; University of South Carolina, Columbia, South Carolina, United States of America

* arebar@mailbox.sc.edu

## Abstract

Fly-in fly-out (FIFO) work involves commuting long distances and living in provided accommodation for 1–4 weeks while on shift. Little is known about couple communication and relationship satisfaction of this population. Separate cohorts of FIFO workers and partners of FIFO workers completed daily surveys to self-report time spent communicating with their partner (in person, over the computer, and by phone) and relationship satisfaction for up to 7 days during on-shift and 7 days during off-shift periods (806 observations from $N$ = 106 with 19 couples). For FIFO workers, but not partners of FIFO workers, relationship satisfaction was lower during on-shift days than off-shift days. This difference was completely mediated by time spent communicating with romantic partner, such that after accounting for the impact of lower time spent communicating on relationship satisfaction, there remained no significant difference in relationship satisfaction on-shift vs off-shift. Communication between FIFO workers and their romantic partners is essential to ensure the relationship does not suffer while the worker is away from home. FIFO organisations need to investigate ways to ensure quality communication between romantic partners can be maintained while the worker is on-site and partner spend long periods of time away from one another.

## Introduction

Around 20% of regionally-based male Australians are employed in 'Fly-in, Fly-out' (FIFO; or 'drive-in, drive-out') posts in regional Australia, most commonly in mining and construction sectors [1]. FIFO workers travel long distances to the worksite and live in nearby communal camps during their on-shift roster of 1–4 weeks in a row of typically 12-hour workdays [2]. FIFO work has been described as having several benefits, most notably the higher income compared to the median personal income

**Data availability statement:** All data and script are freely available at https://osf.io/xw52s/?view_only=55782e4f040a404586d91a-c5e7266e6f.

**Funding:** The not-for-profit organisation LIVIN Australia donated $250 worth of incentives (hats) for participants in the study, but had no influence on the study procedures, data collection, analyses or dissemination. (No grant number or award received).

**Competing interests:** The authors have declared that no competing interests exist.

in Australia [3]. FIFO work enables employees and their families to have flexibility to reside in a place of their preference, despite the worker's regularly changing worksite locations of typically regional and remote areas [4]. Some have pointed to mental health benefits of FIFO work, including being part of a challenging work environment and access to unique opportunities to experience new locations, and meet new people [2,5]. Alongside the noted benefits of FIFO work, there are also several potential disadvantages of this style of employment, such as inadequate sleep and poorer psychological distress compared to the average Australian worker [6–9]. There are also potential negative consequences on FIFO workers relationships, given that they are separated from their partner and/or family for the length of their on-shift roster times.

The transient nature of FIFO work means that workers and their partners live separately for 1-to-4 successive weeks [10]. Physical distance between partners in a relationship can reduce emotional intimacy and make it difficult to navigate the roles that each member plays in the relationship [11–12]. Not surprisingly then, the distance between the worksite and home residency can leave some FIFO workers feeling displaced from family, friends, and social networks [6]. FIFO workers also describe psychological and physical distance as a source of tension [11–12].

FIFO workers and partners routinely must cope with fluctuating between a long-distance relationship and being close to their partner. The issues that arise in FIFO relationship may mirror those of other forms of long-distance relationships including more likelihood of jealousy, and less relational trust compared to geographically close relationships [13]. Individuals in long-distance relationships tend to experience elevated stress both, within and outside of the relationship, compared to geographically close couples [14]. Couples in long-distance relationships can find relationship satisfaction difficult to maintain as a result of the lack of consistency in proximity between partners [15]. Little, however, remains known about relationship satisfaction of FIFO workers and partners of FIFO workers and whether perceptions of relationship satisfaction differ between when the FIFO worker is on-shift vs off-shift.

### The importance of time spent communicating between FIFO workers and partners

Spending time communicating with one's partner is a vital element in the maintenance of a healthy relationship [16–17]. Social interactions between partners including even 'basic communication' like small talk are instrumental for positive relationship satisfaction [18]. Perhaps consequently, couples in long-distance relationships face challenges communicating and remaining emotionally connected. The regular disruptions to physical proximity and long workdays of FIFO work means FIFO workers and their partners have less face-to-face interaction time, limited opportunities for daily conversation, less shared free time, and less physical intimacy [19].

For FIFO workers and their partners, there is a cyclical pattern of being in a long-distance relationship while the worker is on-shift and then being together during off-shift time. During the on-shift roster phase, partners will likely spend less time

communicating together given that the worker typically has long workdays that can vary between day or night shifts [10]. Little is understood about how this unique work shift impacts communication and relationship satisfaction of FIFO workers and their partners. Understanding whether there is an impact on communication and relationship satisfaction for workers and partners is imperative for the wellbeing of FIFO workers and their families [2]. A repeated measures study, in which variables are assessed daily, is needed for a comprehensive understanding of how relationship satisfaction changes and the systematic influences on this important aspect of overall well-being.

### The present study

The aim of this study was to test whether there are differences in daily relationship satisfaction for separate cohorts of FIFO workers and partners of FIFO workers when the FIFO worker is on-shift (away from home) and off-shift (at home with partner), and whether any impact can be explained through time partners spend communicating. It was hypothesised that relationship satisfaction for both worker and partner would be worse on on-shift vs off-shift days. It was also hypothesised that the difference in relationship satisfaction between on- and off-shift times would be partially or fully mediated by time spent communicating with partner.

## Materials & methods

### Study procedures

The study was conducted as part of the FIFO Lifestyle Project – a repeated assessment study in which separate cohorts of FIFO workers and partners of FIFO workers were invited to participate in a study about the physical, mental, and social impacts of FIFO work. Participants were recruited through social media forums, and traditional media outlets (e.g., newspaper, radio, and news items) and directed to an online study website through which they provided informed consent by agreeing via an electronic selection and responded to eligibility screening questions. To be eligible, participants had to be 18 years or older and self-identify as a FIFO worker or a partner of a FIFO worker, with the description of a FIFO worker provided (i.e., FIFO workers travel long distances to the worksite and live in nearby communal camps during their on-shift roster of 1–4 weeks in a row of typically 12-hour workdays). Participants were also asked if they would like to invite their partner to the study. This choice was optional and not inviting their partner did not exclude them from study involvement. If participants did opt in to inviting their partner, the partner's email address was collected and research staff sent a study invite email. In return for their involvement, participants were entered into a random draw for $30 AUD (US$24) gift vouchers, a value which we deemed to be motivating, but not coercive, for potential participants.

Eligible participants provided contact details and self-reported demographic information including their sex, age, and postcode. Participants were then contacted by research staff to schedule daily surveys based on their upcoming work roster. Participants were sent up to 7 consecutive on-shift days and 7 consecutive off-shift days (i.e., total of 14 surveys) at 16:00 with links to secure web-based surveys assessing relationship satisfaction and time spent communicating with partners. The emails instructed participants to complete the surveys at the end of the day. For workers with 14 day shifts, surveys the mid-point of FIFO workers' next on-shift period (days 4–11) and for those with 7-day shifts received surveys occurred across all 7 days. Data were collected between January 2017 and May 2017. All study procedures were approved by the Central Queensland University Human Research Ethics Committee (Ethics Number: H16/09–269).

### Participants

Prior to the study being conducted, an a priori sample size estimation was calculated to inform our target sample size of 100 people with 6 observations each to achieve power of 80% to find medium effect sizes for fixed effects [20]. Recruitment was stopped once the target sample size was achieved. In total, there were 806 daily reports from 64 FIFO workers and 42 partners of FIFO workers. Within the dataset, there were 19 couples, with the remaining portion of the sample

consisting of reporting on partners who did not participate themselves in the study. FIFO workers were predominantly male (51 men, 13 women) aged 40.49 ± 10.34 years old. The most commonly reported occupations were managers/professionals (n=30, 28.3%), tradespersons/labourers (n=23, 21.7%), production/ transport (n=19, 17.9%) and clerical/sales/service (n=13, 12.3%). Partners of FIFO workers were predominantly female (40 women, 2 no response) aged 38.5 ± 9.22 years old. Household incomes were between $A47 500 and $A350 000 (~US$37 600–US$277 000), with mean earnings $A190 000±$A68 000 (~US$150 500±US$54 000). All participants were based in Australia with most based in Queensland (*n* = 61, 57.5%) and Western Australia (*n* = 24, 22.6%).

### Measures

Relationship satisfaction was measured via a single item ("are you satisfied with your relationship with your partner today?"). Responses ranged from 0 (*no, not at all*) to 3 (*yes, definitely*). Given the high participant burden of full scales in daily surveys, a full scale was not utilised; however, this single item was adapted from an existing valid and reliable relationship scale [21]. The wording of the item was the same as that of the validated scale with the exception that we added 'with your partner' and 'today' to clarify that our focus was on their romantic partner and referred to the time boundaries of that day. Time spent communicating with partner was measured as the time spent communicating in person, over the computer, and by phone, with their partner. Responses were given in hours and minutes and converted into minutes for analyses.

### Data management & analyses

Intraclass correlations (ICCs) and 95% confidence intervals (CIs) were calculated to determine the degree to which relationship satisfaction and time spent communicating with partner varied at the within-person level (commonality of responses of same person over time) and at the within-couple level (commonality of responses between partners within couple) [22]. The hypotheses were tested using evaluation of direct, indirect, and total effects within multilevel models with random effects [23]. The multilevel modelling accounted for three-levels of nesting – within-couple and within-person over time using the *lme4* [23–25] package of *R* 3.6.2 [26]. The mediation models consisted of three models: The first model included relationship satisfaction as a dependent variable and the predictors (i.e., independent variables) of a nominal dichotomized variable of whether it was an on-shift or off-shift day (on-shift/off-shift), a nominal dichotomized variable of whether the participant was a Fly-In, Fly-Out worker or a partner of a Fly-In, Fly-Out worker (worker/partner), and the interaction term between on-shift/off-shift and worker/partner. The second model included the dependent variable of time spent communicating and the same set of predictors. The third model included the dependent variable of relationship satisfaction and predictors of on-shift/off-shift, worker/partner, the interaction term between on-shift/off-shift and worker/partner and time spent communicating with partner. Prior to model estimation, it was confirmed that there were no assumptions violated of non-linearity, multicollinearity, or homoscedasticity. Data and script are all available at https://osf.io/xw52s/?view_only=55782e4f040a404586d91ac5e7266e6f.

### Results

On average, participants completed 7.46 ± 4.89 of 14 daily surveys, with 3.22 ± 2.96 of a possible 7 on-shift surveys, and 4.21 ± 3.84 of 7 possible off-shift surveys. Descriptive statistics are shown in Table 1. On average, people spent about 4.5 hours together per day (with wide variability) and reported a mid-range relationship satisfaction (2.15 on a 0–3 scale). ICCs revealed that about half of the variability in relationship satisfaction was at the person-level with the majority of that explained within-partner. That is, most of the variability of the relationship satisfaction variable was the result of differences between partners as opposed to changes between days. Comparatively, there was far more variability between days in time spent communicating with partner, with less than a quarter of variability shared between-person or between-couple.

## Relationship satisfaction

Table 2 shows the model testing whether relationship satisfaction differed between FIFO workers and FIFO partners, and between on-shift and off-shift days. There was a significant moderation effect found such that the impact of on-shift vs off-shift days differed significantly between workers and partners. This moderation is depicted in Fig 1. Notably, for FIFO workers, there is a significant drop in relationship satisfaction on on-shift days compared to off-shift days, whereas this effect is not present for partners, and trends in the opposite direction.

**Table 1.** *Descriptive statistics of relationship satisfaction and time spent communicating with partner.*

| Variable | Mean (SD) | Range | Skewness | Kurtosis | Person level (over time) ICC (95% CI) | Couple level (within partners) ICC (95% CI) |
|---|---|---|---|---|---|---|
| 1. Relationship Satisfaction | 2.15 (0.96) | 0–3 | -0.85 | -0.35 | 0.51 (0.43 to 0.59) | 0.26 (0.14 to 0.48) |
| 2. Time Spent Communicating with Partner | 278.55 (397.15) | 0–1440 | 1.70 | 1.96 | 0.16 (0.10 to 0.24) | 0.09 (0.02 to 0.25) |

806 observations from *N* = 105 with 19 couples

ICC: Intraclass Correlation

**Table 2.** **Results of model testing whether daily relationship satisfaction differed between Fly-in, Fly-out workers and partners and between on-shift vs off-shift days.**

| Dependent Variable: Relationship Satisfaction | *b* | 95% Confidence Interval |
|---|---|---|
| Fixed Effects | | |
| Intercept | 2.05* | 1.81 to 2.29 |
| On-Shift vs Off-Shift | 0.13 | -0.09 to 0.34 |
| Worker vs Partner | 0.16 | -0.13 to 0.45 |
| On-Shift vs Off-Shift × Worker vs Partner | -0.32* | -0.57 to -0.06 |
| Random Effects | *Variance* | *SD* |
| Intercept | 0.15 | 0.39 |
| Couple | 0.40 | 0.63 |
| Residual | 0.45 | 0.67 |

806 observations from *N* = 106 with 19 couples

*p <.05

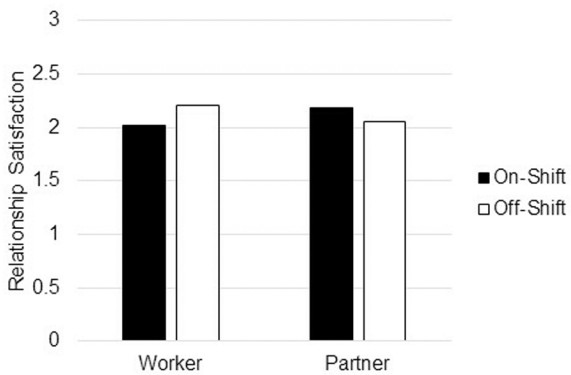

**Fig 1. Relationship satisfaction for Fly-in, Fly-out workers and partners of Fly-in, Fly-out workers during on-shift vs off-shift days.**

### Time spent communicating with partner

The model testing difference of time spent communicating with partner between on-shift and off-shift days and between workers and partners is shown in Table 3. Similar to the findings of relationship satisfaction, there was a significant moderation effect found such that the impact of on-shift vs off-shift days differed significantly between workers and partners. This moderation is depicted in Fig 2. Not surprisingly, both workers and partners reported much more time spent communicating with partners on off-shift days compared to on-shift days. Notably, however, for workers, there was a much larger difference in reported time spent communicating with partner between on-shift and off-shift days than for partners.

### Time spent communicating with partner explains difference in relationship satisfaction between on-shift and off-shift days

The third model, which is the last step of the mediation analyses, revealed that once the time spent communicating with partner was included in the model, the systematic difference in relationship satisfaction between on-site and off-site

**Table 3. Results of model testing whether time spent communicating with romantic partner differed between Fly-in, Fly-out workers and partners and between on-shift vs off-shift days.**

| Dependent Variable: Time Spent Communicating with Romantic Partner | b | 95% Confidence Interval |
|---|---|---|
| Fixed Effects | | |
| Intercept | 348.76* | 282.68 to 414.73 |
| On-Shift vs Off-Shift | -187.36* | -283.63 to -90.43 |
| Worker vs Partner | 129.50* | 42.28 t0 217.20 |
| On-Shift vs Off-Shift × Worker vs Partner | -233.91* | -348.45 to -120.08 |
| Random Effects | *Variance* | *SD* |
| Intercept | 0.13 | 0.12 |
| Couple | 0.40 | 0.62 |
| Residual | 0.46 | 0.66 |

806 observations from *N* = 106 with 19 couples

*p <.05

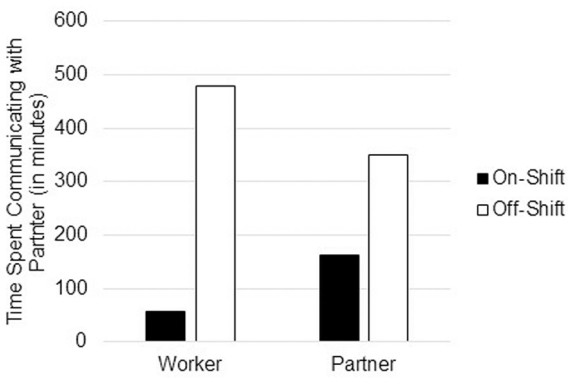

**Fig 2. Reported daily time spent communicating with romantic partner for Fly-in, Fly-out workers and partners of Fly-in, Fly-out workers during on-shift vs off-shift days.**

days was no longer statistically significant (Table 4). The mediation total, direct and indirect effect estimations revealed that time spent communicating fully mediated the difference in relationship satisfaction between on-shift vs off-shift days experienced by FIFO workers, with a total effect of 18.25 (95% CI: 5.67, 30.83), an indirect effect of 22.14 (95% CI: 12.34, 34.36), and a direct effect of -3.89 (95% CI: -14.56, 6.78). The mediation model is presented in Fig 3.

## Discussion

This study aimed to investigate whether being on-site during FIFO work impacted daily relationship satisfaction for separate cohorts of FIFO workers and partners of FIFO workers. Additionally, it was tested whether the time partners spent communicating mitigated differences in daily relationship satisfaction between when the FIFO worker was on-shift vs off-shift. This is the first study to provide this important evidence of the fluctuations in relationship satisfaction and communication between partners across FIFO rosters through a repeated assessment design. Notable also is that, whereas most FIFO work research focuses exclusively on the perspectives of FIFO workers, we assessed perspectives of FIFO workers and partners of FIFO workers. Our hypotheses were partially supported with results indicating that FIFO workers, but not

**Table 4. Results of model testing whether time spent communicating with romantic partner mediated the difference of relationship satisfaction between on-shift vs off-shift days.**

| Dependent Variable: Relationship Satisfaction | b | 95% Confidence Interval |
|---|---|---|
| Fixed Effects | | |
| Intercept | 1.94* | 1.70 to 2.18 |
| On-Shift vs Off-Shift | 0.18 | -0.03 to 0.40 |
| Worker vs Partner | 0.11 | -0.17 to 0.39 |
| On-Shift vs Off-Shift × Worker vs Partner | -0.24 | -0.49 to 0.01 |
| Time Spent Communicating with Partner | < 0.01* | < 0.00 to < 0.00 |
| Random Effects | *Variance* | *SD* |
| Intercept | 0.14 | 0.37 |
| Couple | 0.40 | 0.63 |
| Residual | 0.44 | 0.67 |

806 observations from *N* = 106 with 19 couples

*p <.05

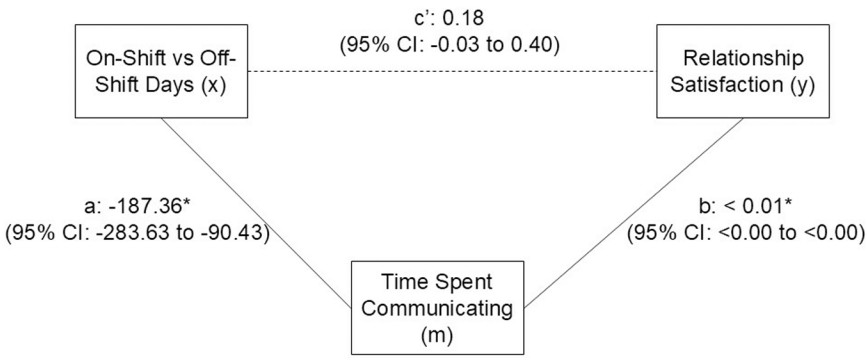

Indirect effect: 22.14 (95% CI: 12.34 to 34.36)

**Fig 3. Mediation model showing daily time spent communicating with romantic partner for Fly-in, Fly-out workers and partners of Fly-in, Fly-out workers mediate the difference in relationship satisfaction between on-shift vs off-shift days.**

partners of FIFO workers, were less satisfied with their relationships on on-shift days compared to off-shift days. This drop in relationship satisfaction during on-shift periods was fully explained by less time spent communicating with romantic partners during on-shift days.

Similar to previous studies investigating dynamics of relationship satisfaction [27–28], the present study findings reveal that relationship satisfaction varies day-to-day. This study extends previous research by investigating relationship satisfaction within the context of the unique structure of FIFO work. Our findings reveal that there is likely an impact on being away from home on FIFO workers' relationship satisfaction. Along with existing evidence of the potential negative impact of FIFO work stressors and circumstances on relationships [8,10,12], this study highlights the needs for FIFO industries to be aware of these potential negative influences of the work style on their employees and consider how to mitigate these potentially detrimental effects.

It was anticipated that relationship satisfaction would be lower for FIFO workers and partners of FIFO workers when workers were on-shift compared to off-shift; however, this effect was only found for the cohort of FIFO workers. Notably, this group difference was also found for perceived time spent communicating with their partner: The cohort of FIFO workers reported larger reductions in time spent communicating with partners when on-shift vs off-shift compared to the cohort of partners of FIFO workers. It is important to consider that the sample consisted of data from a few couples with both romantic partners in the study, but most participants were in relationships with partners that were not participating in the study. Before conclusions can be drawn about whether there are differences within-couples of the perceived impact of FIFO work on relationship satisfaction and communication, studies with larger samples of both partners in a relationship need to be conducted.

Our findings showed that the negative impacts of on-shift vs off-shift roster times on FIFO workers' relationship satisfaction were not seen for days that workers communicated more with their partners. Given the importance of communication for relationship satisfaction [17,18,29] and the evidenced impact of long-distance relationships on communication and relationship satisfaction [14–16], it may be worthwhile for FIFO industries to consider enhancing the means and opportunities workers have to communicate with their partners back home. Further evidence suggests that relationship maintenance training may be a useful tool for supporting the relationship between FIFO workers and their at-home partners [30].

There are multiple avenues for communicating with partners while FIFO workers are at the on-site residence including videoconferencing, messaging, and phone conversations. FIFO organisations should ensure that workers are provided with sufficient resources (i.e., internet connectivity) and time to interact with their partners and family while they are on-site. However, communicating via these means have been shown to have lower psychological and interpersonal outcomes than in person interactions [31]. It may be that worksite visits in times of hardship or flexible allowances for home visits could mitigate some potential negative impacts of being away from family while working on site. Moreover, an increase in mobile phone and Wi-Fi accessibility along with communication training to increase the amount of meaningful communication may be beneficial. Beyond enhancing opportunities to communicate with partners, FIFO organisations may consider aiding their employees in improving the quality of communication with their romantic partners while on-shift [29].

## Study limitations and future directions

Given the daily, repeated nature of the study, it was elected to use a single item measure of relationship satisfaction. It may be that, compared to other options for assessing relationship satisfaction [21,32], these measures had limited validity. Additionally, the self-reported item assessing time spent communicating with partners may reflect some response bias that would have been eliminated if we had captured this data with objective alternatives that were not available to us (e.g., phone records). Furthermore, the causal direction of the effects cannot be tested given the observational nature of the study. We speculated that time spent communicating with romantic partner influenced relationship satisfaction; however, it is also likely that this is a reciprocal relationship such that when people are less satisfied with their relationship, they will

tend to engage in less communication with their romantic partner than when they are more satisfied with their relationship. Future intervention work is needed to tease out the directionality of these effects. Additionally, evidence suggests that the content of communication more so than overall time spent communicating has an impact on relationship satisfaction [33]. Further inquiry into how partners stay connected, and the quality or content of their communication is important to guide intervention or support efforts for FIFO workers and their families. Future research should also consider whether other aspects of relationships (e.g., length of relationship, relationship status) may also play a role in the impact of communication and FIFO work on relationship satisfaction.

This study was a comparison of perceptions of FIFO workers and partners of FIFO workers so it does not provide for conclusions about the impact of FIFO work on relationship satisfaction or communication with romantic partner compared to that of other occupations. Importantly, future work is needed to compare how FIFO workers compare to non-FIFO workers in terms of relationship satisfaction these, and other factors important for successful romantic relationships, between occupations. For example, it would be interesting to tease apart the effects on a relationship of travel, roster length, and occupational stressors of FIFO work [12]. It is worth noting, too, that our sample consisted of FIFO workers who were predominantly male and partners who were predominantly female. More work is needed to determine whether these findings replicate for same sex couples and for which the FIFO worker identifies as female and the partner as male. Also, given that the aim of the study was to consider short-term fluctuations in relationship satisfaction, this study does not provide evidence of the long-term, accumulate consequences of FIFO work on relationships. Further work should consider tracking longer time periods of relationships throughout FIFO work careers, particularly in new FIFO workers which may provide a better sense of the fractionality of effects.

## Conclusions

FIFO work is a prominent career option amongst regional Australians. There are benefits of this style of work including a relatively high income; however, there are potential costs as well including impact on relationships. The present study revealed that relationship satisfaction for FIFO workers declines when they are away from home, living on-site for the on-shift part of their roster. Importantly, we found that when partners spend more time communicating with each other, this negative impact of being away from one another can be mitigated. This study leads to the call for FIFO organisations to support efforts to facilitate communication between FIFO workers and their partners.

## Acknowledgments

None.

## Author contributions

**Conceptualization:** Amanda L. Rebar.

**Data curation:** Amanda L. Rebar.

**Methodology:** Ashlee Forshaw, Kristie-Lee Alfrey.

**Writing – original draft:** Ashlee Forshaw.

**Writing – review & editing:** Corneel Vandelanotte, Benjamin Gardner, Amanda L. Rebar.

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
