## [Decision Letter · Decision Letter 0]

26 Nov 2024

PONE-D-24-43342Communication and Relationship Satisfaction of Fly-In, Fly-Out Workers and PartnersPLOS ONE

Dear Dr. Gardner,

Thank you for submitting your manuscript to PLOS ONE. After careful consideration, we feel that it has merit but does not fully meet PLOS ONE’s publication criteria as it currently stands. Therefore, we invite you to submit a revised version of the manuscript that addresses the points raised during the review process.

Be sure to:

answer on the requests and suggestions of both reviewers regarding statistical analysis an the interpretation of the resultsprovide more characteristics of the data in Supplements

We look forward to receiving your revised manuscript.

Kind regards,

Sanja Batić Očovaj, PhD

Academic Editor

PLOS ONE

Journal Requirements:

Additional Editor Comments:

The problem that you researched is very important and interesting, however, there is a need for the thorough improvement of the article or even the broadening of the sample.

Firstly, the authors should format the article according to Plos One's requests.

The abstract should present information about the sample and instruments. Both reviewers agreed that the sample is too small and that detected relationships are questionable. The sample should be described in the Method. It would be useful to provide additional information about demographic details e.g., ethnicity, type of job, income, relationship length, and relationship status. Plus, the authors should discuss the power of the sample.

The data about relationship satisfaction should be written in the Results in the section Relationship Satisfaction. The data about skewness and kurtosis of the relationship satisfaction and communication time should be presented. Descriptive statistics for both measures should be presented separately for the period on and off-site.

The results should be organized more logically. The contradictions and mistakes in the presented results addressed by reviewers should be avoided and corrected. I.e. saying that relationship satisfaction is regressed against the nominal dichotomized variable of whether it was an on-shift or off-shift day (on-shift/off-shift), the nominal dichotomized variable of whether the participant was a Fly-In, Fly-Out worker or a partner of a Fly-In, Fly-Out worker (worker/partner), and the interaction term between on-shift/off-shift and worker/partner, means that relationship satisfaction is a proposed dependent variable. The finding that workers spend more time communicating with their partners than their partners with them is questionable. The presented results of the mediation analysis should be checked and rewritten.

After addressing the problematic issues the authors should make necessary corrections in the discussion.

Reviewers' comments:

Reviewer's Responses to Questions

**Comments to the Author**

1. Is the manuscript technically sound, and do the data support the conclusions?

Reviewer #1: Yes

Reviewer #2: Partly

2. Has the statistical analysis been performed appropriately and rigorously? 

Reviewer #1: Yes

Reviewer #2: Yes

3. Have the authors made all data underlying the findings in their manuscript fully available?

Reviewer #1: Yes

Reviewer #2: Yes

4. Is the manuscript presented in an intelligible fashion and written in standard English?

Reviewer #1: Yes

Reviewer #2: Yes

5. Review Comments to the Author

Reviewer #1: This scholarly article is written in plain language, its statistical analyses are acceptable and it is novel and original, I appreciated getting to learn about the Australian FIFO population.

How do the authors account for the impact of testing timing factors upon degree of satisfaction? Relatedly, does having only a single-item measure not intrinsically limit the validity of any conceptual inference at all when its examination is indicated to a time period for participants, let alone when it comes to an intersubjective conceptual inference whereby an Other is posited? I think a work-completion and relaxation, diurnal and/or if not a stress effect is likely here and might more properly have been controlled for, but the size of the effect would be doubtful due to missing and limited statistical power.

Is there a possible regression to the mean effect involving general feelings about the relationship from a partner within the actual investigated experimental manipulation based on the poor completion rate of 7.46 ± 4.89 of 14 daily surveys?

“Notably, however, for workers, there was a much larger difference in reported time spent communicating with partner between on-shift and off-shift days than for partners.” Is this not entirely attributable to the effect of stress, travel or both?

“All hypotheses were supported with results indicating that FIFO workers, but not partners of FIFO workers, were less satisfied with their relationships on on-shift days compared to off-shift days.” Is it not tautological, at worst a post hoc ergo propter hoc fallacy, to evaluate the impact of specialized work on one set of workers but not for another, even independent set of workers (ie the partners) while investigating their shared feelings for a common point of reference between the two that is presumably partially or entirely affected by the work? I would want FIFO workers to be compared to other experimental groups such as temporary workers, etc.

Reviewer #2: I reviewed this article for another journal before. The authors have made no changes since then. Thus, I have copied and pasted my original review below.

Thank you for the opportunity to review the manuscript entitled “Time Spent Communicating Between Fly-In, Fly-Out Workers and Partners and its Impact on Relationship Satisfaction” (the title is different in the journal system than in the manuscript). I really wanted to like this paper because I found the sample studied very interesting (I will soon be in a FIFO relationship myself and know many people in this situation) as there aren’t very many studies looking at these types of relationship specifically. There is more research into long-distance relationships more generally but not this subgroup specifically. I can imagine the sample was also quite difficult to recruit and hard to get people to complete the surveys when they’re doing 12-hour shifts. However, this means that the sample size is very small, and the response rate is also quite low.

It's useful to include the sample size in the abstract.

The data analysis section sounds like the regression is done the wrong way around (relationship satisfaction as a predictor of on-shift/off-shift) but I can see from the data analysis script that it’s done right. I’d reword the section but be clearer.

It doesn’t quite make sense to me to report the average time spent communicating with partner overall. I think it should be split by on/off-site. I think relationship satisfaction also makes sense to split.

Demographic details of the sample are lacking e.g., ethnicity, type of job, income, relationship length, relationship status.

There’s no discussion of power or effect sizes in the study. My guess is that the study is quite underpowered and especially looking at any between-partner effects won’t be very meaningful with only 19 couples participating in the study.

To me it doesn’t quite make sense that the workers report spending more time communicating with their partners than their partners report communicating with them. This is of course because most of the participants did not participate as a couple but to say there’s a difference is non-sensical because if they’re communicating with each other they have to be spending the same amount of time communicating with the other. Unless they’re texting of course, and one person is sending 100 messages and the other 1 and not reading the messages. But this is probably an edge-case.

I do not understand the results from the mediation analyses. These are not reported in Table 4 (I can only see moderation). Also, the confidence intervals of “time spent communicating with partner” do not include the actual beta coefficient. Also, the beta coefficient is so small that I can’t see how this is meaningful and how it’s even significant. Additionally, the authors state that the indirect effect is 0.09 and total effect 8.71. This would suggest that almost all of the total effect is direct effect i.e., there is no mediation at all.

I’m not quite clear from the paper what past research exists on FIFO workers and their partners.

It feels quite misleading to state things like “Notable also is the novelty amongst FIFO work research that we assessed perspectives of FIFO workers and partners of FIFO workers” because the authors only had 19 couples out of the 108 participants.

6. PLOS authors have the option to publish the peer review history of their article (what does this mean? ). If published, this will include your full peer review and any attached files.

**Do you want your identity to be public for this peer review?** For information about this choice, including consent withdrawal, please see our Privacy Policy .

Reviewer #1: **Yes: ** Paul-André Betito, HBA*, MSW, RSW

Reviewer #2: No

---

## [Author Response · Author response to Decision Letter 1]

17 Dec 2024

Response to Reviewers

PONE-D-24-43342

Editor comments:

1. Please provide additional details regarding participant consent. In the ethics statement in the Methods and online submission information, please ensure that you have specified (1) whether consent was informed and (2) what type you obtained (for instance, written or verbal, and if verbal, how it was documented and witnessed). If your study included minors, state whether you obtained consent from parents or guardians. If the need for consent was waived by the ethics committee, please include this information.

We have revised the Methods section to include the requested details regarding informed consent. No minors were included in the study.

Methods: “Participants were recruited through social media forums, and traditional media outlets (e.g., newspaper, radio, and news items) and directed to an online study website through which they provided informed consent by agreeing via an electronic selection and responded to eligibility screening questions.”

2.. Your ethics statement should only appear in the Methods section of your manuscript. If your ethics statement is written in any section besides the Methods, please move it to the Methods section and delete it from any other section. Please ensure that your ethics statement is included in your manuscript, as the ethics statement entered into the online submission form will not be published alongside your manuscript.

We have removed the ethics statement from the Cover Letter.

Additional Editor Comments:

1. The problem that you researched is very important and interesting, however, there is a need for the thorough improvement of the article or even the broadening of the sample.

Firstly, the authors should format the article according to Plos One's requests.

The abstract should present information about the sample and instruments. Both reviewers agreed that the sample is too small and that detected relationships are questionable. The sample should be described in the Method. It would be useful to provide additional information about demographic details e.g., ethnicity, type of job, income, relationship length, and relationship status. Plus, the authors should discuss the power of the sample.

We thank you for noting the importance of the topic and its practical implications. We have revised the manuscript tin line with the Plos One guidelines.

Power and sample size: As detailed in our response to Reviewer 2, comment 6, we have included details about the power of the study and that recruitment of participants was stopped when sufficient power was achieved. We believe that the confusion about power and sample size was due to our lack of clarity that the sample was made of two separate cohorts of participants with a total N of 106 (Cohort 1: 64 FIFO workers, Cohort 2: 42 partners of FIFO workers) rather than only from 19 couples.

Abstract: We have revised the abstract to include more details about the sample and the measures: “Separate cohorts of FIFO workers and partners of FIFO workers completed daily surveys to self-report time spent communicating with their partner (in person, over the computer, and by phone) and relationship satisfaction for up to 7 days during on-shift and 7 days during off-shift periods (806 observations from N = 106 with 19 couples).”

Sample characteristics. We moved the description of our sample to the Methods section. Additionally, as detailed in our response to Reviewer 2, comment 5, we have included more details of the sample characteristics in our revised manuscript including details of household income and FIFO worker occupations.

2. The data about relationship satisfaction should be written in the Results in the section Relationship Satisfaction. The data about skewness and kurtosis of the relationship satisfaction and communication time should be presented. Descriptive statistics for both measures should be presented separately for the period on and off-site.

We have included the skewness and kurtosis of the variables in Table 1. Given that we depict the differences in relationship satisfaction and time spent communicating as a function of both on-shift vs off-shift days and between FIFO workers and partners of FIFO workers in the Figures, we elected to not duplicate this in the Table. If the editor feels strongly that this revision is essential, we can make the revision.

3. The results should be organized more logically. The contradictions and mistakes in the presented results addressed by reviewers should be avoided and corrected. I.e. saying that relationship satisfaction is regressed against the nominal dichotomized variable of whether it was an on-shift or off-shift day (on-shift/off-shift), the nominal dichotomized variable of whether the participant was a Fly-In, Fly-Out worker or a partner of a Fly-In, Fly-Out worker (worker/partner), and the interaction term between on-shift/off-shift and worker/partner, means that relationship satisfaction is a proposed dependent variable.

We have revised the terminology as requested. Please see response to Reviewer 2’s comment 3 regarding the changes made.

4. The finding that workers spend more time communicating with their partners than their partners with them is questionable. The presented results of the mediation analysis should be checked and rewritten.

We have provided clarity on these points and made revisions to the manuscript based on these Reviewer comments. Please see our responses to Reviewer 1 Comment 3 as well as to Reviewer 2 Comment 7 for our responses regarding the difference in perceived time spent communicating with romantic partners. Please see our response for Reviewer 2 Comment 8 for our response regarding the revision of the mediation analyses and results.

5. After addressing the problematic issues the authors should make necessary corrections in the discussion.

We have made several revisions to the Discussion as referred to in the relevant comment responses including referring to updated evidence on the topic, which is detailed in our response to Reviewer 2, comment 9.

Reviewer's Comments

Reviewer #1: This scholarly article is written in plain language, its statistical analyses are acceptable and it is novel and original, I appreciated getting to learn about the Australian FIFO population.

We appreciate the comment.

1. How do the authors account for the impact of testing timing factors upon degree of satisfaction? Relatedly, does having only a single-item measure not intrinsically limit the validity of any conceptual inference at all when its examination is indicated to a time period for participants, let alone when it comes to an intersubjective conceptual inference whereby an Other is posited? I think a work-completion and relaxation, diurnal and/or if not a stress effect is likely here and might more properly have been controlled for, but the size of the effect would be doubtful due to missing and limited statistical power.

As is standard for repeated measures studies in which participants repeatedly complete the same set of measures over time, we intentionally specified the measure wording to be specific to the construct on that day (e.g., “are you satisfied with your relationship with your partner today?”). This approach allows for consideration of how much the variable varies day-to-day and for capturing the impact of fluctuations in relationships between variables at the daily level (Shiffman et al., 2008). That the item is a single item may limit the interitem reliability of the scale in a single assessment, but is common and best practice for repeated measures studies such as this one, in which case the balance must be met with interitem reliability and participant burden and fatigue. We have made it clear in the revision that the self-report measures is a subjective one of the participant’s perspective on their relationship rather than an objective assessment of the relationship, especially (as noted by the reviewer) that under most circumstances, this is self-reported by only one of the two members of the relationship.

Shiffman, S., Stone, A. A., & Hufford, M. R. (2008). Ecological momentary assessment. Annual Review of Clinical Psychology, 4(1), 1-32. https://doi.org/10.1146/annurev.clinpsy.3.022806.091415

Discussion: “Given the daily, repeated nature of the study, it was elected to use single item measures of relationship satisfaction and time spent communicating with partners. It may be that, compared to other options for assessing relationship satisfaction [21,32], these measures had limited validity.”

Note also that the study was sufficiently powered and the recruitment stopped when sufficient power was achieved. We have revised the Methods section to include these details.

Methods: “Prior to the study being conducted, an a priori sample size estimation was calculated to inform our target sample size of 100 people with 6 observations each to achieve power of 80% to find medium effect sizes for fixed effects [20].”

2. Is there a possible regression to the mean effect involving general feelings about the relationship from a partner within the actual investigated experimental manipulation based on the poor completion rate of 7.46 ± 4.89 of 14 daily surveys?

To clarify, our study was observational with no experimental manipulation. Additionally, note that the number of responses received was largely dependent on the participants’ work rosters rather than non-compliance. For example, many participants did not work for 7 days consecutively so were not eligible for some of the on-site surveys. Also, given the study has been properly powered, there is minimal to null risk that the completion rate would have an influence on findings. Regarding the point about regression to the mean, we would argue that because ours is a repeated measures study, the risk of findings being impacted by regression to the mean is far less than is typical for studies with a single or a few measures. Rather, we can capture fluctuations around the mean (both the sample mean and the individual-level mean). Our findings test systematic links between fluctuations of the variables and compare those associations between people.

3. “Notably, however, for workers, there was a much larger difference in reported time spent communicating with partner between on-shift and off-shift days than for partners.” Is this not entirely attributable to the effect of stress, travel or both?

We appreciate the point. To highlight this important point, we included mention in the revised Discussion that further work is needed to further tease out which aspects of FIFO work are driving these effects.

Discussion: “This study was a comparison of perceptions of FIFO workers and partners of FIFO workers so it does not provide for conclusions about the impact of FIFO work on relationship satisfaction or communication with romantic partner compared to that of other occupations. Importantly, future work is needed to compare these, and other factors important for successful romantic relationships, between occupations. For example, it would be interesting to tease apart the effects on a relationship of travel, roster length, and occupational stressors of FIFO work [12].”

4. All hypotheses were supported with results indicating that FIFO workers, but not partners of FIFO workers, were less satisfied with their relationships on on-shift days compared to off-shift days.” Is it not tautological, at worst a post hoc ergo propter hoc fallacy, to evaluate the impact of specialized work on one set of workers but not for another, even independent set of workers (ie the partners) while investigating their shared feelings for a common point of reference between the two that is presumably partially or entirely affected by the work? I would want FIFO workers to be compared to other experimental groups such as temporary workers, etc.

We appreciate the point. Our research aim was to focus on the perceptions of relationship of FIFO workers and partners of FIFO workers. We were not interested in comparing the impact of FIFO work on relationship satisfaction to that of other occupations, although that is a worthwhile avenue for future research. To highlight this important point, we included mention in the revised Discussion that further work is needed to determine if this is a phenomenon specific to FIFO work or is generalizable to other occupations (including those that travel vs those who do not).

Discussion: “This study was a comparison of perceptions of FIFO workers and partners of FIFO workers so it does not provide for conclusions about the impact of FIFO work on relationship satisfaction or communication with romantic partner compared to that of other occupations. Importantly, future work is needed to compare these, and other factors important for successful romantic relationships, between occupations. For example, it would be interesting to tease apart the effects on a relationship of travel, roster length, and occupational stressors of FIFO work [12].”

Reviewer #2: I reviewed this article for another journal before. The authors have made no changes since then. Thus, I have copied and pasted my original review below.

1. Thank you for the opportunity to review the manuscript entitled “Time Spent Communicating Between Fly-In, Fly-Out Workers and Partners and its Impact on Relationship Satisfaction” (the title is different in the journal system than in the manuscript). I really wanted to like this paper because I found the sample studied very interesting (I will soon be in a FIFO relationship myself and know many people in this situation) as there aren’t very many studies looking at these types of relationship specifically. There is more research into long-distance relationships more generally but not this subgroup specifically. I can imagine the sample was also quite difficult to recruit and hard to get people to complete the surveys when they’re doing 12-hour shifts. However, this means that the sample size is very small, and the response rate is also quite low.

We appreciate your interest in the study. Indeed, it is a unique, understudied population. However, the study was not underpowered (please see responses to comment 6). Note also that the response rate is largely dependent on the work roster of the participants (many did not have 7 consecutive on-site days) rather than non-compliance.

As the findings of our study and others indicate, the relationship of a FIFO worker and their partner can be a rewarding but difficult one to manage, mainly as a result of the shifting between two ‘types of lifestyles’. Our interest in the population originated as a result of the experiences of a member of our research team as a partner of FIFO worker, so we appreciate the complexity of the situation and wish you and your partner the best – make sure to communicate with each other, especially while they are on-site.

2. It's useful to include the sample size in the abstract.

We have added the sample size to the abstract: “Separate cohorts of FIFO workers and partners of FIFO workers completed daily surveys to self-report time spent communicating with their partner and relationship satisfaction for up to 7 days during on-shift and 7 days during off-shift periods (806 observations from N = 106 with 19 couples).”

3. The data analysis section sounds like the regression is done the wrong way around (relationship satisfaction as a predictor of on-shift/off-shift) but I can see from the data analysis script that it’s done right. I’d reword the section but be clearer.

Within statistical language the term ‘variable X regressed onto Y’ refers to variable X as the outcome, and Y as a set of predictors. However, we recognise the problem and have revised the language to be clearer for Plos One readership.

Methods: “The hypotheses were tested using evaluation of direct, indirect, and total effects within multilevel models with random effects [23]. The multilevel modelling accounted for three-levels of nesting – within-couple and within-person over time using the lme4 [23-25] package of R 3.6.2 [26]. The mediation models consisted of three models: The first model included relationship satisfaction as a dependent variable and the predictors (i.e., independent variable

---

## [Decision Letter · Decision Letter 1]

30 Jan 2025

PONE-D-24-43342R1Communication and relationship satisfaction of fly-in, fly-out workers and partnersPLOS ONE

Dear Dr. Gardner,

Thank you for submitting your manuscript to PLOS ONE. After careful consideration, we feel that it has merit but does not fully meet PLOS ONE’s publication criteria as it currently stands. Therefore, we invite you to submit a revised version of the manuscript that addresses the points raised during the review process.

**ACADEMIC EDITOR: The article is significantly improved. It is very interesting and all reviewers are very interested in the subject.  In order to additionaly improve the article so additional revievers were engaged and they had some useful comments so the authors should put some additional effort to to improve the text according to their suggestions. It is required that authors **

more clearly state the outline of the study or its primary contribution,add required information about the used single item measure of Relationship Satisfaction and a comment about its psychometric characteristics in  Method, show the plot for mediational analysis  in the section Results,enrich the Discussion with more thorough questionning of the meaning of the results regarding of the possible impact demographic characteristics of partners, the quality of communication, type and duration of the relationships, reformulate the conclusion s regarding the Hypothesis as reviewer 4 suggested,comment about the analysis of demographic factors as  control variables or additional mediators of relationship satisfaction in the analysis,explain why they did not reported and commented random effects.

We look forward to receiving your revised manuscript.

Kind regards,

Sanja Batić Očovaj, PhD

Academic Editor

PLOS ONE

Journal Requirements:

Reviewers' comments:

Reviewer's Responses to Questions

**Comments to the Author**

1. If the authors have adequately addressed your comments raised in a previous round of review and you feel that this manuscript is now acceptable for publication, you may indicate that here to bypass the “Comments to the Author” section, enter your conflict of interest statement in the “Confidential to Editor” section, and submit your "Accept" recommendation.

Reviewer #1: All comments have been addressed

Reviewer #3: (No Response)

Reviewer #4: (No Response)

Reviewer #5: All comments have been addressed

Reviewer #6: All comments have been addressed

2. Is the manuscript technically sound, and do the data support the conclusions?

Reviewer #1: Yes

Reviewer #3: Yes

Reviewer #4: Yes

Reviewer #5: Yes

Reviewer #6: Yes

3. Has the statistical analysis been performed appropriately and rigorously? 

Reviewer #1: Yes

Reviewer #3: Yes

Reviewer #4: Yes

Reviewer #5: Yes

Reviewer #6: Yes

4. Have the authors made all data underlying the findings in their manuscript fully available?

Reviewer #1: Yes

Reviewer #3: Yes

Reviewer #4: Yes

Reviewer #5: Yes

Reviewer #6: Yes

5. Is the manuscript presented in an intelligible fashion and written in standard English?

Reviewer #1: Yes

Reviewer #3: Yes

Reviewer #4: Yes

Reviewer #5: Yes

Reviewer #6: Yes

6. Review Comments to the Author

Reviewer #1: I thank the authors for their systematic treatment of the comments and feedback provided, their scholarly article remains appropriate for publication despite its conventional and expectable limitations.

The authors have done a good job of implementing feedback, especially of highlighting the applicability and presentation of basic statistical principles to/in their analyses, something I think was necessary given the most dubious aspect of the article, its methodology in relation to its sampling. I have no further critical comments to suggest.

I also thank the authors for their input about and reinforcement of ecological momentary assessment.

Reviewer #3: Thank you for submitting your manuscript to the Journal. The updated version of the manuscript is appreciable.

I have thoroughly reviewed it and found it appropriate for publication after incorporating changes suggested in the 1st stage of the review process. However, after clarity on the part of sample size and sample cohorts, as the paper established that it was “Separate cohorts of FIFO workers and partners of FIFO workers” and not all of them were couples it creates a strong need for understanding of several other contingencies and boundary conditions that may cause the variation in response because the two cohorts are dissimilar in all settings except the factor under study. It may offer several future directions of research.

Reviewer #4: Thank you for the opportunity to review the manuscript titled: “Communication and relationship satisfaction of fly-in, fly-out workers and partners”. This manuscript examines FIFO workers and FIFO partners’ self-reported relationship satisfaction and time spent communicating when working away from home (on-shift) and at home (off-shift). As expressed by the other reviewers, I enjoyed learning about the FIFO workers and FIFO partners and the authors should be commended for conducting research on an underrepresented population. Overall, I think the manuscript has promise but there are a few outstanding issues that, if addressed, I believe would improve the work. I have listed these below.

Major points:

1. The introduction provides no outline of the study or its primary contribution. It would be helpful for a reader to have a paragraph (or even a couple of sentences) outlining what the study examines and its main contribution. Currently, the introduction reads more like a literature review.

2. Some more information about the study design is required in the methods. As one example, from the previous response to reviewers, I can understand the reason for using a single item to measure relationship satisfaction. However, in what ways was the item “adapted from an existing valid and reliable relationship scale” (p. 7)? Was it taken verbatim or was it rewritten? If the latter, what was the justification? As another example, I wasn’t sure what were the benefits of using daily surveys? This isn’t to say the design is inappropriate, but rather that it requires some justification.

3. It would have been interesting to have additional mediators of relationship satisfaction in the analysis (e.g., relationship status), but I understand that these were not gathered for the study. However, I see that some demographic details were collected that may influence relationship satisfaction (notably Age and Gender). What were the reasons for not including these in the models?

4. Was there a reason for not reporting the random effects (ID & Relationship ID) in the results (i.e., in Tables 2,3, and 4)?

5. There is missing some discussion on the validity of the findings given that they were derived exclusively from self-reported scales, rather than direct observations of behaviour. I understand the study is looking into the participant’s perspectives of relationship satisfaction and time spent communicating, however I think some more discussion is required about what this signifies. For instance, it is difficult to interpret the significance of the reported time spent communicating – how should organizations or policy makers act on this information given it might be a product of different perspectives on time spent rather than actual differences in time spent?

6. There is a statement in the discussion about hypotheses being supported in the Discussion that doesn’t seem accurate to the findings:

“It was hypothesised that relationship satisfaction for both worker and partner would be worse on on-shift vs off-shift days.” (Present Study, p. 5)

“All hypotheses were supported with results indicating that FIFO workers, but not partners of FIFO workers, were less satisfied with their relationships on on-shift days compared to off-shift days” (Discussion, p. 13)

In my understanding, the hypothesis stated was not supported as only relationship satisfaction for FIFO workers was lower for on-shift vs off-shift days (Fig 1), while the opposite was found for FIFO partners. Indeed, the authors themselves state that:

“It was anticipated that relationship satisfaction would be lower for FIFO workers and partners of FIFO workers when workers were on-shift compared to off-shift; however, this effect was only found for the cohort of FIFO workers.” (p. 13)

Minor points:

1. The sentence below is confusing:

“FIFO workers also report troubles establishing and maintaining longterm relationships and describe psychological and physical distance can be a source of tension in relationships” (p. 3)

Perhaps it should read “…describe how psychological and physical distance can be a source of” or “describe psychological and physical distance as a source of tension”

2. There is a typo on p. 4: “more likelihood of jealously” should be “jealousy”

3. This sentence is confusing “Notable also is the novelty amongst FIFO work research that we assessed perspectives of FIFO workers and partners of FIFO workers.” (p. 13)

Do the authors mean that it there is a notable absence of prior working researching FIFO workers and partners?

Reviewer #5: Thank you for giving me the opportunity to review this paper on such an interesting topic of fly-in, fly-out works and partners in the Australian context. The results were interesting, showing how partners and workers differ in their relationship satisfaction. Your discussion on how this may be due to differences in the perceptions of communication time or lack of complete couples in the sample shows good response to the previous reviewers.

Based on the revised version, I would accept the manuscript as ready for publication. If the authors do wish to make improvements, I'd consider including mediation diagrams in the main text to make it easier for the readers new to the topic to understand the analysis, although you do provide all necessary data in your tables.

Reviewer #6: Dear editors,

Thank you for the opportunity to review the article titled "Communication and relationship satisfaction of fly-in, fly-out workers and partners". I have read the original version and the updated version with the additions and changes suggested by the other two reviewers. I believe that the manuscript has improved significantly in the new version and the missing and unclear parts have been adequately addressed. Considering the few studies that currently exist on this topic, I think the present study is very interesting, despite its limitations (which were well pointed out by the authors).

I have no major revisions to suggest, just a few minor ones.

- Regarding the sample, it is not clear from the description of the recruitment whether participants received compensation for their participation in the study.

- An element that I believe was not considered in the study, but which could be suggested as a future direction, is also the difference in perception in relation to gender.

- I see that the study was conducted before the COVID period. It would of course have been interesting to compare the period before and after, as the pandemic has exacerbated many factors related to employee wellbeing, including the work-life sphere. However, it might be appropriate to point out that when monitoring and managing long-term Covid symptoms, it becomes even more important to consider the relationship component in this particular category of workers, as this is an important protective factor for maintaining a good work-life balance, which contributes to overall well-being.

7. PLOS authors have the option to publish the peer review history of their article (what does this mean? ). If published, this will include your full peer review and any attached files.

**Do you want your identity to be public for this peer review?** For information about this choice, including consent withdrawal, please see our Privacy Policy .

Reviewer #1: **Yes: ** Paul-André Betito, HBA*, MSW, RSW

Reviewer #3: No

Reviewer #4: No

Reviewer #5: No

Reviewer #6: No

---

## [Author Response · Author response to Decision Letter 2]

18 Feb 2025

Editor:

The article is significantly improved. It is very interesting and all reviewers are very interested in the subject. In order to additionally improve the article so additional reviewers were engaged and they had some useful comments so the authors should put some additional effort to to improve the text according to their suggestions. It is required that authors:

1. more clearly state the outline of the study or its primary contribution,

2. add required information about the used single item measure of Relationship Satisfaction and a comment about its psychometric characteristics in Method,

3. show the plot for mediational analysis in the section Results,

4. enrich the Discussion with more thorough questionning of the meaning of the results regarding of the possible impact demographic characteristics of partners, the quality of communication, type and duration of the relationships,

5. reformulate the conclusion s regarding the Hypothesis as reviewer 4 suggested,

6. comment about the analysis of demographic factors as control variables or additional mediators of relationship satisfaction in the analysis,

7. explain why they did not reported and commented random effects.

RESPONSE TO EDITOR COMMENTS:

Thank you for your efforts to support the peer-review of this manuscript, which we agree reports an important finding with implications for FIFO workers, their partners, and FIFO organizations. In line with recommendations from you and the reviewers, we have made these additional revisions to the manuscript: We have provided more insights into the contributions of the study in the introduction (Please see response to Reviewer 4, comment 1), details about the measure of relationship satisfaction (Please see response to Reviewer 4, comment 2), included a mediation diagram as a new Figure (Please see response to Reviewer 5, comment 1), reformulated the conclusions about the hypotheses being partially supported instead of fully supported (Please see our response to Reviewer 4, comment 6), added further discussion about the consideration of demographic factors and other aspects of inter-relationship dynamics in future research (Please see our responses to Reviewer 3, comment 1 and Reviewer 4, comment 3), and reported the random effects for all models (Please see response to Reviewer 4, comment 4).

Reviewer #3:

I have thoroughly reviewed it and found it appropriate for publication after incorporating changes suggested in the 1st stage of the review process. However, after clarity on the part of sample size and sample cohorts, as the paper established that it was “Separate cohorts of FIFO workers and partners of FIFO workers” and not all of them were couples it creates a strong need for understanding of several other contingencies and boundary conditions that may cause the variation in response because the two cohorts are dissimilar in all settings except the factor under study. It may offer several future directions of research.

OUR RESPONSE: We appreciate the comments and agree that further work needs to be done to understand the inter-relationship dynamics of navigating FIFO work, as is noted in our Discussion: “It is important to consider that the sample consisted of data from a few couples with both romantic partners in the study, but most participants were in relationships with partners that were not participating in the study. Before conclusions can be drawn about whether there are differences within-couples of the perceived impact of FIFO work on relationship satisfaction and communication, studies with larger samples of both partners in a relationship need to be conducted.”

Reviewer #4:

1. The introduction provides no outline of the study or its primary contribution. It would be helpful for a reader to have a paragraph (or even a couple of sentences) outlining what the study examines and its main contribution. Currently, the introduction reads more like a literature review.

OUR RESPONSE: The “The Present Study” paragraph section is intended to provide context around the study aims and its contribution. Beyond this, we have included more in the introduction regarding the study design and its contribution to advancing our understanding of this important issue for FIFO workers and their partners:

“Understanding whether there is an impact on communication and relationship satisfaction for workers and partners is imperative for the wellbeing of FIFO workers and their families [2]. A repeated measures study, in which variables are assessed daily, is needed for a comprehensive understanding of how relationship satisfaction changes and the systematic influences on this important aspect of overall well-being.”

“The present study: The aim of this study was to test whether there are differences in daily relationship satisfaction for separate cohorts of FIFO workers and partners of FIFO workers when the FIFO worker is on-shift (away from home) and off-shift (at home with partner), and whether any impact can be explained through time partners spend communicating. It was hypothesised that relationship satisfaction for both worker and partner would be worse on on-shift vs off-shift days. It was also hypothesised that the difference in relationship satisfaction between on- and off-shift times would be partially or fully mediated by time spent communicating with partner.”

2. Some more information about the study design is required in the methods. As one example, from the previous response to reviewers, I can understand the reason for using a single item to measure relationship satisfaction. However, in what ways was the item “adapted from an existing valid and reliable relationship scale” (p. 7)? Was it taken verbatim or was it rewritten? If the latter, what was the justification? As another example, I wasn’t sure what were the benefits of using daily surveys? This isn’t to say the design is inappropriate, but rather that it requires some justification.

OUR RESPONSE: We included more details about the purposeful adaptation of the item: “The wording of the item was the same as that of the validated scale with the exception that we added ‘with your partner’ and ‘today’ to clarify that our focus was on their romantic partner and referred to the time boundaries of that day.”

This insight into changes in relationship satisfaction is unique to this field but is a common approach for understanding variables that change across time. That relationship satisfaction was shown to change day-to-day and be systematically linked to a person’s daily life context (on-shift vs off-shift) was, we’d argue, a unique strength of the study. We have included a sentence in the introduction to further justify the study design: “A repeated measures study, in which variables are assessed daily, is needed for a comprehensive understanding of how relationship satisfaction changes and the systematic influences on this important aspect of overall well-being.”

3. It would have been interesting to have additional mediators of relationship satisfaction in the analysis (e.g., relationship status), but I understand that these were not gathered for the study. However, I see that some demographic details were collected that may influence relationship satisfaction (notably Age and Gender). What were the reasons for not including these in the models?

OUR RESPONSE: Our sample had very limited heterogeneity in terms of age and gender, with 95% of FIFO worker partners identifying as female and 75% of the sample 35-45 years old. So, we agree that these are demographic characteristics that may impact these findings, we are not able to conduct these analyses in a way that generalizes to the target population.

4. Was there a reason for not reporting the random effects (ID & Relationship ID) in the results (i.e., in Tables 2,3, and 4)?

OUR RESPONSE: We had omitted reporting the random effects for parsimony of reporting of results, but have now included them for all models reported in Tables 2-4.

5. There is missing some discussion on the validity of the findings given that they were derived exclusively from self-reported scales, rather than direct observations of behaviour. I understand the study is looking into the participant’s perspectives of relationship satisfaction and time spent communicating, however I think some more discussion is required about what this signifies. For instance, it is difficult to interpret the significance of the reported time spent communicating – how should organizations or policy makers act on this information given it might be a product of different perspectives on time spent rather than actual differences in time spent?

OUR RESPONSE: We appreciate the point and have included further discussion on the limitation of this measure: “Additionally, the self-reported item assessing time spent communicating with partners may reflect some response bias that would have been eliminated if we had captured this data with objective alternatives that were not available to us (e.g., phone records).” Given the added participant burden and risk of privacy breeches involved with attaining communication records of partners, we’d argue that the benefits of gaining a more objective measure of communication does not outweigh the risks and costs.

6. There is a statement in the discussion about hypotheses being supported in the Discussion that doesn’t seem accurate to the findings: “It was hypothesised that relationship satisfaction for both worker and partner would be worse on on-shift vs off-shift days.” (Present Study, p. 5) “All hypotheses were supported with results indicating that FIFO workers, but not partners of FIFO workers, were less satisfied with their relationships on on-shift days compared to off-shift days” (Discussion, p. 13). In my understanding, the hypothesis stated was not supported as only relationship satisfaction for FIFO workers was lower for on-shift vs off-shift days (Fig 1), while the opposite was found for FIFO partners. Indeed, the authors themselves state that: “It was anticipated that relationship satisfaction would be lower for FIFO workers and partners of FIFO workers when workers were on-shift compared to off-shift; however, this effect was only found for the cohort of FIFO workers.” (p. 13).

OUR RESPONSE: Our apologies for this oversight. Indeed, the findings only partially aligned with our hypotheses. We have revised the statement to clarify this: “Our hypotheses were partially supported with results indicating that FIFO workers, but not partners of FIFO workers, were less satisfied with their relationships on on-shift days compared to off-shift days.”

Minor points:

7. The sentence below is confusing: “FIFO workers also report troubles establishing and maintaining longterm relationships and describe psychological and physical distance can be a source of tension in relationships” (p. 3). Perhaps it should read “…describe how psychological and physical distance can be a source of” or “describe psychological and physical distance as a source of tension”

OUR RESPONSE: Thank you for the great recommendation: “FIFO workers also describe psychological and physical distance as a source of tension [11-12].”

8. There is a typo on p. 4: “more likelihood of jealously” should be “jealousy”

OUR RESPONSE: Thank you. We have corrected the mistake.

9. This sentence is confusing “Notable also is the novelty amongst FIFO work research that we assessed perspectives of FIFO workers and partners of FIFO workers.” (p. 13). Do the authors mean that it there is a notable absence of prior working researching FIFO workers and partners?

OUR RESPONSE: We revised the sentence for clarity: “Notable also is that, whereas most FIFO work research focuses exclusively on the perspectives of FIFO workers, we assessed perspectives of FIFO workers and partners of FIFO workers.”

Reviewer #5:

1. Based on the revised version, I would accept the manuscript as ready for publication. If the authors do wish to make improvements, I'd consider including mediation diagrams in the main text to make it easier for the readers new to the topic to understand the analysis, although you do provide all necessary data in your tables.

OUR RESPONSE: Thank you for the comments. We have included the Mediation diagram as Fig. 3 in the revised manuscript.

Reviewer #6:

1. Regarding the sample, it is not clear from the description of the recruitment whether participants received compensation for their participation in the study.

OUR RESPONSE: We have included these details in the revised manuscript: “In return for their involvement, participants were entered into a random draw for $30 AUD (US$24) gift vouchers, a value which we deemed to be motivating, but not coercive, for potential participants.”

2. An element that I believe was not considered in the study, but which could be suggested as a future direction, is also the difference in perception in relation to gender.

OUR RESPONSE: Great point. We have discussed this in the revised manuscript: “It is worth noting, too, that our sample consisted of FIFO workers who were predominantly male and partners who were predominantly female. More work is needed to determine whether these findings replicate for same sex couples and for which the FIFO worker is female and the partner is male.”

3. I see that the study was conducted before the COVID period. It would of course have been interesting to compare the period before and after, as the pandemic has exacerbated many factors related to employee wellbeing, including the work-life sphere. However, it might be appropriate to point out that when monitoring and managing long-term Covid symptoms, it becomes even more important to consider the relationship component in this particular category of workers, as this is an important protective factor for maintaining a good work-life balance, which contributes to overall well-being.

OUR RESPONSE: We considered adding this as a limitation to our study but ultimately chose not to make changes here. We understand the reviewer’s perspective, but the same could be said for coping with any long-term health condition that may emerge for the FIFO worker or their partner. Indeed, the same could be said for any major life event, whether health-related or not (e.g., coping with bereavement). Given that we did not account for long Covid, or for any external health or other long-term events that may occur during FIFO work periods, a sentence speculating as to whether the management of long-term Covid symptoms might affect relationships would not sit well in this paper.

---

## [Decision Letter · Decision Letter 2]

21 Mar 2025

Communication and relationship satisfaction of fly-in, fly-out workers and partners

PONE-D-24-43342R2

Dear Dr. Gardner,

We’re pleased to inform you that your manuscript has been judged scientifically suitable for publication and will be formally accepted for publication once it meets all outstanding technical requirements.

Kind regards,

Sanja Batić Očovaj, PhD

Academic Editor

PLOS ONE

Additional Editor Comments (optional):

Reviewers' comments:

Reviewer's Responses to Questions

**Comments to the Author**

1. If the authors have adequately addressed your comments raised in a previous round of review and you feel that this manuscript is now acceptable for publication, you may indicate that here to bypass the “Comments to the Author” section, enter your conflict of interest statement in the “Confidential to Editor” section, and submit your "Accept" recommendation.

Reviewer #4: (No Response)

Reviewer #5: All comments have been addressed

2. Is the manuscript technically sound, and do the data support the conclusions?

Reviewer #4: Yes

Reviewer #5: Yes

3. Has the statistical analysis been performed appropriately and rigorously? 

Reviewer #4: Yes

Reviewer #5: Yes

4. Have the authors made all data underlying the findings in their manuscript fully available?

Reviewer #4: Yes

Reviewer #5: Yes

5. Is the manuscript presented in an intelligible fashion and written in standard English?

Reviewer #4: Yes

Reviewer #5: Yes

6. Review Comments to the Author

Reviewer #4: Thank you for the thoughtful responses to my comments and I am satisfied with how these were addressed. I appreciated the additional clarity about how the single-item measure of relationship satisfaction was altered, the additional reporting of Random Effects in Tables 2 and 3, and the addition of the mediation diagram (Fig. 3), as suggested by Reviewer #5. These changes have substantially improved the manuscript. I am therefore happy to recommend the manuscript for publication. My only comment is that I found a typo in Table 3 p. 11 “42.28 t0 217.20” should be “42.28 to 217.20”.

Reviewer #5: All my comments were addressed and the authors have worked hard to improve on the paper. Congratulations on the new version.

7. PLOS authors have the option to publish the peer review history of their article (what does this mean? ). If published, this will include your full peer review and any attached files.

**Do you want your identity to be public for this peer review?** For information about this choice, including consent withdrawal, please see our Privacy Policy .

Reviewer #4: No

Reviewer #5: No

---

## [Editor Report · Acceptance letter]

PONE-D-24-43342R2

PLOS ONE

Dear Dr. Gardner,

I'm pleased to inform you that your manuscript has been deemed suitable for publication in PLOS ONE. Congratulations! Your manuscript is now being handed over to our production team.

Kind regards,

on behalf of

Dr. Sanja Batić Očovaj

Academic Editor

PLOS ONE